# Fractographic and Microhardness Evaluation of All-Ceramic Hot-Pressed and CAD/CAM Restorations after Hydrothermal Aging

**DOI:** 10.3390/ma15113987

**Published:** 2022-06-03

**Authors:** Roxana Diana Vasiliu, Ion-Dragoș Uțu, Lucian Rusu, Adrian Boloș, Liliana Porojan

**Affiliations:** 1Center for Advanced Technologies in Dental Prosthodontics, Department of Dental Prostheses Technology (Dental Technology), Faculty of Dental Medicine, “Victor Babeș” University of Medicine and Pharmacy Timișoara, Eftimie Murgu Sq. No. 2, 300041 Timișoara, Romania; roxana.vasiliu@umft.ro; 2Departament of Materials Engineering and Fabrication, University of Politehnică Timişoara, Bd. Mihai Viteazul nr.1, 300222 Timişoara, Romania; dragos.utu@upt.ro; 3Department Mechanics and Vibrations, Faculty of Mechanical Engineering, Politehnica University, 1 Mihai Viteazu Street, 300222 Timisoara, Romania; lucian.rusu@upt.ro; 4Department of Oral Rehabilitation (Dental Technique), Faculty of Dental Medicine, “Victor Babeș” University of Medicine and Pharmacy Timișoara, Eftimie Murgu Sq. No. 2, 300041 Timișoara, Romania; bolos.adrian@umft.ro

**Keywords:** all-ceramic glass restorations, static loading test, stereomicroscope, fractographic failure analysis

## Abstract

All-ceramic dental restorations have great advantages, such as highly esthetical properties, a less complex fabrication, and a similar abrasion resistance to enamel. Despite these advantages, ceramic materials are more prone to fracture due to their brittle microstructure. The main aim of this in vitro study was to determine the difference in hot-pressed and milled glass-ceramic mechanical properties such as fracture resistance and microhardness (VHN). Four types of ceramics, two hot-pressed and two milled, feldspathic glass-ceramics and zirconia-reinforced glass-ceramics were selected in this study and tested using the static loading test and Vicker’s testing. Hydrothermal aging, consisting of different baths with temperatures between 5 degrees Celsius and 55 degrees Celsius, was chosen as the in vitro aging method. Statistical analyses are performed using SPSS Statistics software at a significance level of *p* < 0.05. Micro-hardness values decrease after hydrothermal aging. The static loading test reveals a significant difference between the feldspathic hot-pressed glass-ceramic, which fractures at lower forces, and milled zirconia-reinforced lithium silicate glass-ceramic, which fractures at greater forces (N). Fractographic analysis of the fractured fragments resulted in the static loading test revealing different surface features about the crack origins and propagations under a stereomicroscope.

## 1. Introduction

Nowadays, all-ceramic materials are considered to be the most esthetical choice in restorative dentistry [1,2]. Despite the fact that their mechanical structures are constantly bettering, these materials still present a brittle character that makes them prone to fracturing after clinical use.

New studies in the dentistry literature have revealed that the main reason for the failure of dental restorations is their fracture after clinical use [3,4,5,6,7]. These failures of the ceramic crowns can be described as multiple fragments or complete fractures of the crowns, and also the tooth structure [8,9,10,11]. Moreover, dental ceramic materials are exposed to thermal and mechanical factors in the oral cavity.

From a microstructural point of view, ceramics can be classified into several categories and subcategories [12,13,14]. The different characteristics of glass-ceramics in the crystalline phase, such as the size and distribution of the crystals, influence their mechanical behavior. According to the classification [14], low-to-moderate leucite with feldspathic glass is so-called ‘’feldspathic porcelains’’. These materials have indications of anterior dental restorations because of their highly aesthetical appearance and their low mechanical strength [15]. 

A new class of improved glass-ceramics that strives to combine mechanical strength and aesthetical properties is zirconia-reinforced lithium silicate (ZLS). This type of glass-ceramic microstructure is based on a lithium-metasilicate glass-ceramic, reinforced with 10% nanoparticles of zirconium dioxide [16]. ZLS glass-ceramic is part of the hybrid ceramic types that combine both the aesthetical properties of glass-ceramic and the mechanical resistance of zirconia [17]. Studies have shown that, when comparing ZLS and feldspathic glass-ceramics, the best fatigue behavior was statistically presented by the ZLS ceramic because of its reinforced microstructure [18]. Since their development, several studies have researched ZLS and feldspathic glass-ceramics milled or hot-pressed [19,20,21,22,23,24,25].

Feldspathic glass-ceramic and ZLS glass-ceramic have clinical indications for monolithic dental restorations. These ceramics can be processed through hot-pressing or milling technology from CAD/CAM blocks. The hot-pressing technology is a laboratory method that enables final dental restorations with less porosity using industrially fabricated ceramic ingots [26]. With hot-pressing technology, both temperature and pressure are applied to prefabricated ceramic ingots, which melt and fill a mold.

In the CAD/CAM technologies, all-ceramic dental restorations are milled industrially from partially or fully sintered prefabricated blocks. ZLS glass-ceramic requires additional sintering, also called crystallization, in order to achieve its fully crystallized form (FCs). A similar study aimed to compare ZLS glass-ceramic with lithium disilicate glass-ceramic based on their crystallization processes and concluded that higher mechanical strength was provided by the ZLS glass-ceramic [25]. 

In the research area, there are several loading devices that are used to study the fracture strength of dental restorations. The mechanical resistance of dental ceramic restorations can be evaluated using tests with specific loads. One of the methods is static loading until the ceramic crowns fracture. The tested crowns are cemented on abutments and placed in a testing machine, where a stylus applies force until the crown breaks [27,28,29,30,31]. The fractographic analysis includes the examination of the resulted fragments and reveals the fracture origin and its path [32,33]. The essential tool in performing the fractography of the failed results is the stereomicroscope. This equipment is used to visualize the samples in 3D and to observe the spatial relationship of the fragments [11,28,34]. This method uses different illumination angles and is able to provide information about the surface roughness of the examined sample [35]. Studies state that the main cause of the failure of ceramic crowns is a radial crack that propagates through the entire material, thus producing the final bulk fracture [36,37]. Chipping of the ceramic material can also be seen in this type of restoration. Some studies suggest that the chipping of all-ceramic crowns can have reasons such as areas of porosity, overloading, or other surface defects [38,39,40]. Fractographic features such as hackles, wake hackles, and arrest lines are indicators of the origin of the fractures [11,41].

Vicker’s microhardness testing is also a frequently used method for mechanical property testing. This method uses an indenter that is pressed at a specific load into a surface. The indenter force is maintained for 10 s. The resulting indentation is microscopically analyzed [42].

Although in vitro studies lack the clinical component, the oral environment can be reproduced to some extent with hydrothermal aging. This method consists of standardized thermal baths with temperatures of 5 degrees Celsius and 55 degrees Celsius for a number of cycles. A series of literature studies [43,44,45] conclude that 10,000 cycles are similar to a year of clinical wear, based on the idea that dental restorations are subject to 20 changes in temperature per day [46,47,48].

This in vitro study aimed to investigate the differences between hot-pressed and milled, feldspathic and zirconia-reinforced lithium silicate (ZLS) glass-ceramics after fractography analysis and microhardness testing. 

The object of this in vitro study was to compare ZLS ceramic with feldspathic glass-ceramic from a mechanical point of view. 

The primary null hypothesis is that there will be no significant difference between the milled and hot-pressed samples regarding the fracture resistance after thermal aging.

The second null hypothesis is that there will be no significant difference regarding the hot-pressed and milled microhardness before and after thermal aging.

## 2. Materials and Methods

In total, 32 all-ceramic crowns were obtained. Half of them were obtained using hot-pressing techniques and the other half were milled using CAD/CAM technology (Table 1). These materials were selected based on their chemical composition as follows: feldspathic glass-ceramic (FP-Vita PM9, Vita Zahnfabrick, Bad Säckingen, Germany), zirconia-reinforced lithium silicate glass-ceramic (ZLSP-Celtra Press, Degudent, Dentsply, Bensheim, Germany), processed using laboratory hot-pressing technology and feldspathic glass-ceramic (FM, Vita Mark II, Vita Zahnfabrick, Germany), and zirconia-reinforced lithium silicate glass-ceramic (ZLSM, Vita Suprinity, Vita Zahnfabrick, Germany), processed by CAD/CAM milling. These materials were included in this research based on their microstructure and their different processing technologies.

### 2.1. Abutment Preparation

The abutment was an upper premolar typodont tooth. It was prepared with a 6° convergence of the axial wall, the reduction of the axial and of the occlusal surface was 1.5 mm. The marginal preparation design was a 1 mm circumferential rounded chamfer.

For standardization the abutment preparation was digitally scanned using a D2000 3D optical scanner (3Shape, Copenhagen, Denmark). After the scanning the prepared abutment, 32 identical resin (Freeprint Model 2.0, Detax GmbH & Co., Ettlingen, Germany) abutment teeth were printed.

### 2.2. CAD/CAM Dental Restorations Fabrication

Sixteen crowns were milled from specific ceramic blocks. In total, 8 ceramic crowns (*n* = 8) were milled from ZLSM blocks and 8 all-ceramic crowns from feldspathic glass-ceramic blocks (FM) (Vita Suprinity; Vita Mark II, Vita Zahnfabrick, Germany).

Pre-sintered ZLSM crowns required a crystallization step after milling. Each specimen was crystalized in the ceramic furnace at a final temperature of 850 °C for about 25 min.

For the microhardness testing using the CAD/CAM blocks were sliced into rectangular-shaped plates (*n* = 32) per material with a thickness of 1.5 mm using a machine (Orthoflex PI Dental, Budapest, Hungary) that provides millimeter accuracy. The rectangular-shaped samples were polished using silicon carbide papers (600–2000 grit) and the final thickness of each specimen was checked with a caliper. The specimens were manually polished with a low-speed handpiece and diamond polishing paste, Renfert Polish (Renfert, Hilzingen, Germany).

Specimen surfaces for CAD/CAM crowns and rectangular-shaped samples were cleaned with alcohol and were glazed.

The two thin layers of the specific glaze were applied manually with a soft brush and fired at the specific parameters respecting the producer’s specification. The two-thin layers of glaze were applied to simulate the same protocol that takes place when cementing the crowns in the oral cavity. For microhardness testing the rectangular-shaped samples were glazed as well to be able to simulate the same protocol as for the crowns (Table 2).

### 2.3. Hot-Pressing Samples Fabrication

In total, 16 crown wax patterns were milled from white Ceramill Wax (Amann Girrbach AG, Kobalch, Östereich), sprued, and invested in a phosphate-bonded investment material (Bellavest SH; BEGO GmbH & Co. KG, Bremen, Germany). Molds were heated at 900 °C for 60 min, and prefabricated ingots (Celtra Press, Dentsply, Degudent; Vita PM9, Vita Zahnfabrick, Bad Säckingen, Germany) were hot-pressed into ceramic specimens using a press furnace (Multimat 2 Touch+ Press Dentsply; Salzburg, Osterreich), following the manufacturer’s guidelines (Table 3). The mold was let to cool down for several hours, and the specimens were carefully devested by sandblasting with glass powders (50 μm) at a pressure of 4 bar. The sprues were cut, and samples prepared for glazing. Two layers of glaze were applied on each sample.

For the microhardness testing, using hot-pressing technique, 32 rectangular-shaped samples were obtained with a thickness of 1.5 mm evaluated using a digital caliper. The specimens were manually polished with a low-speed handpiece and diamond polishing paste, Renfert Polish (Renfert, Hilzingen, Germany). Specimen surfaces for both crowns and rectangular-shaped samples were glazed. The same specific glaze was applied manually with a soft brush and fired at the specific parameters.

### 2.4. Adhesive Placement of the Restorations

All-ceramic crowns were cemented on the abutments using a resin cement, self-etch dual-cure (Maxcem Elite, Kerr, Orange, CA, USA) respecting the producer’s instructions. Before adhesively luting the crowns, the surfaces were etched with 5% hydrofluoric acid. The resin cement was distributed equally on the inner surface of the ceramic crown. The operations were carried out by only one person to ensure the same conditions for every sample.

### 2.5. Hydrothermal Aging of the Samples

Thermocycling was conducted for 10,000 cycles with thermocycling equipment (Thermocycler, SD Mechatronik, Feldkirchen-Westerham, Germany) in two distilled baths with 5 degrees Celsius and 55 degrees Celsius temperatures. The dwelling time was 20 s. The thermal aging process was carried out for all the samples before the mechanical testing for the crowns and the rectangular samples.

### 2.6. Static Loading Test

To investigate the fracture strength, the resin abutments with the ceramic crowns cemented were fixed in the universal testing machine (Instron 3366, Instron Corp, Norwood, MA, USA) using a metal holder that was positioned vertically at 0 degrees to the long axis of the tooth. A stainless-steel stylus of 15 cm with a rounded 6 mm tip was used to express the force. The force was applied on the occlusal surface to simulate the forces during functions such as mastication and chewing at a crosshead speed of 0.5 mm/1 mm (Figure 1). The fracture load was automatically recorded by the equipment at the moment where the fracture of the specimens occurred.

### 2.7. Fractographic Failure Analysis

After the static loading test, all failed specimens were analyzed under a stereomicroscope (Wild M3Z, Heerbrug, Switzerland) to identify the initial crack direction. The analysis begins at one margin of the fractured crown, moving to the occlusal section, and finishing at the other corner of the crown. Equipment magnification ranged from 10 to 100× depending on the analyzed mark seen by constantly moving the angle of the illumination for a better view. Based on the microscope findings a final map was created to indicate the crack propagation for each ceramic crown.

### 2.8. Vickers Microhardness Testing

Samples were evaluated for the Vicker’s microhardness before and after aging. Measurements were made on selected points using the digital camera of the tester, with the microhardness tester DM 8/DM 2 (Yang Yi Technology Co., Ltd, Tainan City 70960, Taiwan) using a diamond pyramidal indenter with 300 g load for 10 s. After lifting the indenter, the indentation dimensions were microscopically recorded (40× magnifications). Five measurements were performed on each surface, and mean values were calculated.

### 2.9. Statistical Analysis

Analyzing the data was performed using SPSS (SPSS version 26.0, IBM, Inc., Chicago, IL, USA). The conducted tests were two-way ANOVA test for the static loading test and the analysis of covariance for the microhardness values. A significance level of ᾳ = 0.05 was set for comparison between the groups. The power of the statistical test was calculated with the same software. Sample size was calculated using specific software (G*Power software 3.1.9.4 (University Kiel, Kiel, Germany). The effect size was chosen 0.50. The calculation revealed 8 samples for each group.

## 3. Results

### 3.1. Fracture Resistance for the Cemented Crowns

The mean values of maximal compressive and displacement and for the hot-pressed and milled crowns are presented in Figure 2 and Figure 3.

The mean deformation load in each group was 312 N for the FP, 438 N for FM, 826 N for ZLSP, and 874,3 N for ZLSM samples. The initial deformation load was 70N for FP, 73.2 for FM, 85.5 for ZLSP, and 87.6 for ZLSM samples. Hot-pressed ceramic crowns exhibited a lower initial deformation load. The differences between the F and ZLS materials remained.

Regarding the processing technology (hot-pressed or milled), significant differences (*p* < 0.001) were found between the values of FP and FM and no significant differences between the samples ZLSP and ZLSM. Both F and ZLS materials reacted the same despite the processing technology, and what the difference was made only by the microstructure.

Regarding the ceramic type, there were significant differences (*p* < 0.05) between F and ZLS ceramics in both groups, hot-pressed and milled for the compressive values during the static loading test. Between the hot-pressed and milled samples, there were no significant differences (*p* > 0.05).

Greater values were found in the ZLSP and ZLSM samples compared with the FM and FP samples (Figure 3).

### 3.2. Vickers Microhardness Evaluation

The samples were tested before and after thermal aging to analyze if their mechanical behavior changed after 10,000 cycles. The microhardness was determined on 64 (n = 16) ceramic samples. The mean values for the tested samples are displayed in Table 4.

With regard to the materials, there was a significant difference between the hot-pressed and the milled samples’ microhardness (*p* < 0.05) before thermal aging. The milled samples feldspathic and ZLS glass-ceramic proved to have greater mean values compared to the hot-pressed samples.

Thermal aging affected the microhardness values of the four materials, with statistically significant changes (*p* < 0.5). The FP ceramic presented the lower microhardness, and the ZLSM ceramic the greater. The ZLSP ceramic was situated between the ZLSM and FM glass-ceramic, with a mean value of 730 N after thermal aging. 

### 3.3. Fractographic Failure Analysis

All the crowns exhibited critical fracture. The initial point was at the contact with or in the proximity of the indenter. The indenter was placed on the occlusal table of the crown. Visible lines of fracture propagated through the entire ceramic material. The cracks also affected the adhesive layer and the resin abutment in some cases. In order to be able to identify the initial site of the crack and its propagation path, fractographic analysis was used. The crack origin for the four types of all-ceramic crowns was located in the contact area of the occlusal edge. Arrest lines are common fracture features that are also good indicators of the direction of the propagation, and they can be seen in Figure 4a. Hackles and edge chips can be seen in Figure 4b,c. Feldspathic glass-ceramic FP exhibited multiple cracks during the static loading test and can be observed in Figure 4d. The ZLSM crown just split into two distinct halves, with the origin of the crack initiated on the occlusal table.

The number of fragments depended on the type of ceramic material and the indenter force. The examined fragments exhibited wake hackles and arrest lines when examined using a stereomicroscope.

## 4. Discussion

The first and second null hypotheses were declared rejected, as significant differences were observed between the ceramic crowns after the static loading test and microhardness testing. Feldspathic glass-ceramic restorations have an indication in the anterior and posterior regions due to their high aesthetics and brittle character. On the other hand, ZLS glass-ceramics display higher mechanical resistance and lower aesthetics. ZLS glass-ceramic has an indication on the posterior areas due to their reinforced microstructure. This study examined all-ceramic anatomical premolar crowns. The crowns were obtained using hot-pressing and milling glass-ceramic. These are the main technologies used in dental practices and laboratories. Based on the results, after the interpretation of the static loading test, the initial deformation load took place at 70 N for FP, 73.2 for FM, 85.5 for ZLSP, and 87.6 for ZLSM. Physiological forces that take place during mastication and deglutition values are between 90–370 N in the anterior teeth and 200–900 N for the posterior teeth [51,52,53]. The masticatory forces may vary depending on the patient’s age, sex, and parafunctions [54]. The materials included in this study are fit for the anterior or posterior region of the oral cavity based on the results.

In vitro studies that include multiple investigations such as hydro-thermal aging, static loading, and microhardness testing are of great importance in testing the materials’ mechanical behavior [55]. In this study, the following two types of materials were compared: a feldspathic glass-ceramic and a reinforced glass-ceramic, both processed through hot-pressing and milling. Several factors can influence the clinical outcome, such as the abutment preparation design, the type of glass-ceramics, thermal aging, and luting agents [56]. In dental practice, ceramic restorations cannot perfectly reproduce and adapt to the marginal preparation, and adhesive luting cements can fill the gaps between the abutment and marginal fit [57,58,59]. Another important aspect is the abutment design, which was chosen carefully based on the materials producer’s instructions because the tooth morphology and geometry have a great impact on the final results [45,60,61]. In the posterior regions of the oral cavity, it is vital to design restorations with simplified occlusal tables or to lower the inclination of the crowns. This aspect was taken into consideration when designing the preparation. The static loading test has revealed that ZLS samples fractured at higher forces (N), compared to the F ceramic, and this makes them fit for the posterior region.

Stereo-microscopy analysis of the fractographic fragments provided a clear identification of the crack features. The chip shapes found on the distal margin of the crown indicate the force application and direction. In Figure 4e, the direction of the chip is parallel to the photographed area. The approach of failure analysis using fractography is able to interpret failure data for all the ceramic restorations. The ceramic crowns exhibited similar fracture features such as hackles, arrest lines, and edge chips. Other features such as velocity hackle and shear hackle were not observed in these ceramic crowns. The FP crowns exhibited multiple cracks compared to the other crowns. The fractographic failure analysis showed that the four types of ceramic split vertically when analyzed with stereomicroscopy. The ZLSM and ZLSP restorations had better performance compared to the other materials from a mechanical point of view. The results from this study have a clinical impact on the long-term survival of all-ceramic crowns. The results have proved that ZLS ceramics, both hot-pressed and milled, are a better solution for the posterior region, but the constant improvement of the ceramic also enables feldspathic glass-ceramic to be used in the posterior region with higher aesthetical results.

The micro-hardness study is considered an important investigation for the mechanical behavior of restorative dental ceramic materials as well. The surface hardness is a method of measuring the resistance of materials to indentation controlled by equipment [62,63,64]. Micro-hardness measurement is actually used to determine the abrasiveness of a material. In the literature, several studies concluded that dental glass-ceramic materials can produce high antagonist wear due to their high microhardness [65]. Additionally, in order to be able to compare the mechanical properties of the materials before and after thermal aging, microhardness on disk-shaped samples was included. In order to simulate the oral environment as much as possible, thermal aging for 10,000 cycles was used [45,46], as well as clinically approved protocols [58]. Thermal aging has a significant effect on the micro-hardness, and all the values for the tested materials increased after 10,000 cycles. The significantly lower microhardness after thermal aging for the FP, FM, and ZLSP can be considered an advantage for protecting the opposing teeth. The differences in the static testing and microhardness values in this study were based mainly on the microstructural differences of the tested materials, such as composition and processing technology. The zirconia-reinforced lithium silicate glass-ceramic had greater values for microhardness and fracture toughness due to the 10 percent zirconia, compared to the feldspathic glass-ceramic. The hot-pressed ceramic presented higher values compared to the milled samples, proving that the processing method has a significant effect on the mechanical properties of dental glass-ceramic.

Similar studies [65,66] in the research literature compared ZLS ceramics with different types of glass-ceramics, such as lithium disilicate and hybrid ceramics, from a microstructural and mechanical point of view. Results showed that ZLS ceramic displayed higher mechanical properties compared to other materials. The same results were obtained in this study. The reason for this research was to see to what degree the microstructure of the ceramic influences the mechanical behavior.

The limitations of this study include the fact that this was only an in vitro study. This is the reason why hydro-thermal aging was included, to simulate the oral environment in a manner. Another limitation of this study is the number of materials. Although these materials are not novel, they are still used in the clinical field due to their proven mechanical and optical properties. Further research should be carried out in which other materials, such as lithium disilicate glass-ceramic, hybrid ceramic, and also zirconia, are included. By including different types of ceramic, the research can provide a wider view of the mechanical properties of the materials.

## 5. Conclusions

Within the limitations of the study, the following conclusions can be drawn:(1)Fracture resistance for the heat-pressed cemented crowns is lower compared to the milled ceramics;(2)The ZLS glass-ceramic revealed higher mechanical properties, microhardness, and fracture toughness compared to the F glass-ceramic;(3)Thermal aging had a significant effect on the microhardness values;(4)The ZLSM restorations had better performance compared to the other materials, both hot-pressed and milled.

## Figures and Tables

**Figure 1 materials-15-03987-f001:**
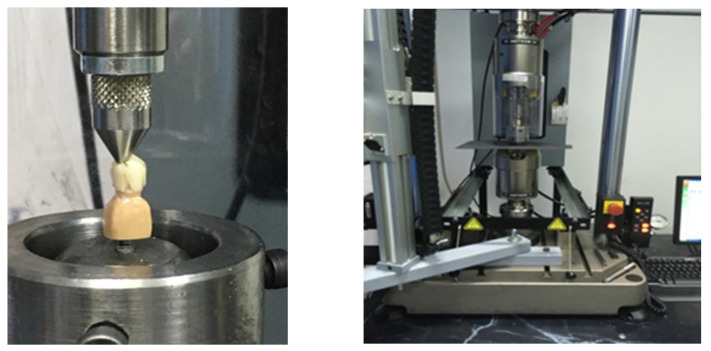
Load-to-fracture test in the universal testing machine before the placement of the tin foil, stating loading test in the universal testing (Instron 3366, Instron Corp, Norwood, MA, USA).

**Figure 2 materials-15-03987-f002:**
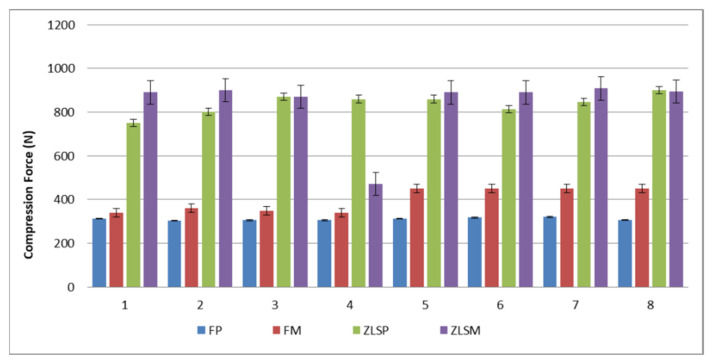
Mean maximal compressive values for the heat-pressed samples FP, ZLSP, and milled FM, ZLSM.

**Figure 3 materials-15-03987-f003:**
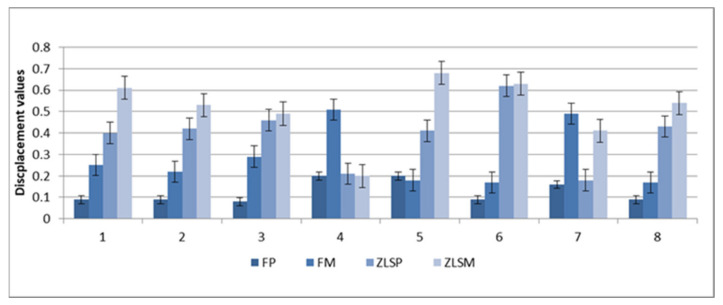
Mean maximal displacement values for the heat-pressed samples FP, ZLSP, and milled FM, ZLSM.

**Figure 4 materials-15-03987-f004:**
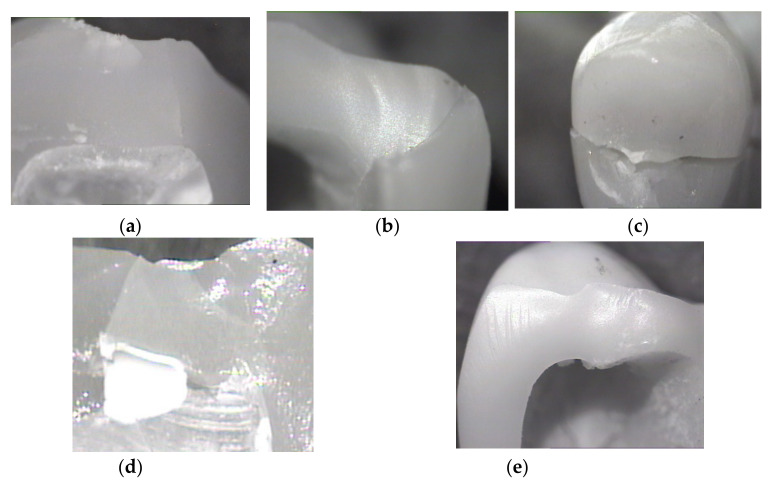
Stereomicroscope images: (**a**) of an occlusal marginal edge of one half of a ZLSP crown, is a clinical view with the origin of the fracture with a hackle and an arrest line. (**b**) Of a distal marginal edge chip of FM crown with the fracture origin and edge chips. (**c**) ZLSM crown split vertically into two halves. (**d**) One half of FP crown with two different directions of crack propagation (**e**) ZLSM and the origin was an edge chip from a force applied on the bottom of the margin aimed upwards, multiple hackles (twit hackles) can be observed as well.

**Table 1 materials-15-03987-t001:** Ceramic dental materials with their composition [49].

Material	Chemical Composition	Manufacturer	Processing Technology
(1) Vita PM9 (FP)	50% (vol%) leucite crystals embedded in the residual glass	Vita Zahnfabrick	Hot-pressing
(2) Celtra Press (ZLSP)	a glass matrix and lithium disilicate crystals having a crystal length of about 1.5 µm plus nano-scale lithium phosphate 10% (ZrO2)	Degudent Dentply	Hot-pressing
(3) Vita Mark II (FM)	<20 wt% feldspathic crystals (average particle size 4 µm) > 80 wt% glass matrix	Vita Zahnfabrick	Milling
(4) Vita Suprinity (ZLSM)	The silica content of 55–65 wt% the lithia (15–21 wt%) zirconia (8–12 wt%) nanoparticle size 0.5–0.7 µm	Vita Zahnfabrick	Milling

**Table 2 materials-15-03987-t002:** Types of glazes used for hot-pressed and milled ceramic [49].

Type of Ceramic	Type of Glaze
FP, FM, ZLSM	Vita Akzent Plus Glaze LT (Vita Zahnfabrick, Bad Sackingen, Germany)
ZLSP	Dentsply Universal Stain (Dentsply, Hanau, Germany)

**Table 3 materials-15-03987-t003:** Parameters for hot-pressing ceramic according to the manufacturer [50].

Parameters	Vita PM9	Celtra Press
Starting temperature	700 °C	700 °C
Hold time	20 min	30 min
Vacuum level	47 hPa	45 hPa
Press time	10 min	3 min
Heat rate	50 °C/min	40 °C/min
Press temperature	1000 °C	860 °C
Press pressure	3 bar	3 bar

**Table 4 materials-15-03987-t004:** Mean values for the microhardness testing. *^, a^ statistically significant.

Samples	Before tc (HVN) (GPa)	After tc (HVN) (GPa)
**FP**	650 *	600
**FM**	680 ^a^	615
**ZLSP**	725 *	690
**ZLSM**	896 ^a^	730

## Data Availability

Not applicable.

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
