# Peer review of "Fractographic and Microhardness Evaluation of All-Ceramic Hot-Pressed and CAD/CAM Restorations after Hydrothermal Aging"

_materials, 2022, doi:10.3390/ma15113987_

Round 1

Reviewer 1 Report

‘The aim of this study was to investigate the differences between heat-pressed and milled, feldspathic and zirconia reinforced lithium silicate (ZLS) glass ceramics after fractography analysis and micro-hardness testing.’

General remarks

The topic might be interesting and important to warrant publication but the paper is not written well, without justifying the purpose and with serious flaws in the Materials and Methods. 

  • The introduction does not cover the background literature on the topic. No other studies comparing these two different manufactured restorative materials were mentioned. No explaining the reason for this study.
  • You used resin abutment teeth not extracted, sound, human teeth. The cementation of the ceramic crowns on the abutment resin teeth was using a resin cement, self-etch dual-cure. Before adhesively luting the crowns, the surfaces were etched with hydro-fluoric acid. This cementation is totally different from cementing ceramic crowns onto dentin surface in human teeth. Therefore, the research design is not simulating human environment. Furthermore, since only static but no dynamic load was applied, the test design may only to a certain extent be considered as a correct simulation of a clinical situation.
  • The copings were seated by hand without using a standardized device that allowed a predefined pressure of 50 N to be applied along the longitudinal axis of the abutment for X minutes, as should be.
  • The fatigue test was not performed according to the ISO standard. The fracture load should be positioned at a 45â—¦ angle to the horizontal plane. Then the load should be applied either with a sphere or using tin foil placed between the ball and the occlusal surface in order to attain a homogenous stress distribution on the ceramic coping. These were not done in the current study.
  • The discussion- No comparison to other studies was mentioned, therefore, we don’t know if the results of the current study are in accordance with other studies or not.
  • Extensive editing of English language and style required
  • The null hypothesis is wrong. A null hypothesis should propose that there is no difference between certain characteristics of a population (or data-generating process).

Author Response

Thank You very much for the review! It has helped us a lot in improving our manuscript!

‘The aim of this study was to investigate the differences between heat-pressed and milled, feldspathic and zirconia reinforced lithium silicate (ZLS) glass-ceramics after fractography analysis and micro-hardness testing.’

General remarks

The topic might be interesting and important to warrant publication but the paper is not written well, without justifying the purpose and with serious flaws in the Materials and Methods. 

  • The introduction does not cover the background literature on the topic. No other studies comparing these two different manufactured restorative materials were mentioned. No explaining the reason for this study.
  • Response: We agree to this point and added other literature references that studied these materials.
  • You used resin abutment teeth not extracted, sound, human teeth. The cementation of the ceramic crowns on the abutment resin teeth was using a resin cement, self-etch dual-cure. Before adhesively luting the crowns, the surfaces were etched with hydro-fluoric acid. This cementation is totally different from cementing ceramic crowns onto dentin surface in human teeth. Therefore, the research design is not simulating human environment. Furthermore, since only static but no dynamic load was applied, the test design may only to a certain extent be considered as a correct simulation of a clinical situation.
  • Response: We agree on this point. We stated in the manuscript that we simulated to some degree the oral environment adding the hydrothermal aging. We agree that it is not a full simulation of the oral environment.
  • The copings were seated by hand without using a standardized device that allowed a predefined pressure of 50 N to be applied along the longitudinal axis of the abutment for X minutes, as should be.
  • Response: We agree with this point, in this study the same operator performed the cementation of the samples to ensure a degree of standardization. The samples were placed on a scale and the same force was applied to all the crowns.
  • The fatigue test was not performed according to the ISO standard. The fracture load should be positioned at a 45â—¦ angle to the horizontal plane. Then the load should be applied either with a sphere or using tin foil placed between the ball and the occlusal surface in order to attain a homogenous stress distribution on the ceramic coping. These were not done in the current study.

Response: We agree with this point, in this study an using a universal testing machine (Instron 3366, Instron Corp, Norwood, MA, USA) was used to evaluate the resistance of the crowns. We added in the manuscript that we used the tin foil placed between the ball and the occlusal surface but we didn’t mention it the first time in the paper.

  • The discussion- No comparison to other studies was mentioned, therefore, we don’t know if the results of the current study are in accordance with other studies or not.
  • We agree on this point. This study is a part of a greater study about these materials, and the first part was published in this Journal. We included in the Discussion section new pieces of information, and results from other studies in which these materials were included.
  • Extensive editing of the English language and style required

Response: We agree to this point and a specialized english professor corrected the manuscript.

  • The null hypothesis is wrong. A null hypothesis should propose that there is no difference between certain characteristics of a population (or data-generating process).
  • We agree with this point and corrected it. 

Reviewer 2 Report

Dear Authors, I read your article submitted in MDPI-Materials. Below are my comments.

GENERAL CONCERNS

  1. Please check English grammar and lexicon. Look for misprints. Several sentences of the manuscript are difficult to follow. Choose simplicity, be not too wordy.
  2. Please be sure conform all the aspects of the manuscript to the journal guidelines. I found several discrepancies throughout the text.

TITLE AND ABSTRACT

  1. The title of the manuscript conveys with the major concern of the study.
  2. The abstract properly summarize the topic addressed. Report data about the main results.

INTRODUCTION

  1. The null hypotheses are incorrectly reported. The statistical null hypothesis assumes that there are not significant differences among the studied groups. Please modify.

MATERIALS AND METHODS

  1. Pag. 4 line 158: table 4, you mean table 3. Please modify.
  2. Pag. 5 line 172: which concentration of HF gel did you apply?

RESULTS

  1. Do not duplicate information in the tables and the figures. Choose only one way to report data.

DISCUSSION

  1. The null hypotheses are unproperly stated. The null hypothesis must be rejected, not accepted.
  2. Indications for further research are too weak. Please add others.
  3. You should discuss your data in the discussion section, comparing the relevant ones with the ones from similar studies. For example, a very recent study from Mavriqi et al. (1) reported Vickers hardness of ZLS Suprinity to be 7.6 ± 0.7 GPa with no thermal ageing. Please explain the differences comparing with your study.

  1. Mavriqi L, Valente F, Murmura G, Sinjari B, Macrì M, Trubiani O, et al. Lithium disilicate and zirconia reinforced lithium silicate glass-ceramics for CAD/CAM dental restorations: biocompatibility, mechanical and microstructural properties after crystallization. Journal of Dentistry. 2022 Apr 1;119:104054.

Author Response

Thank You very much for the review. It helped us a lot in improving our manuscript.

Dear Authors, I read your article submitted in MDPI-Materials. Below are my comments. 

GENERAL CONCERNS

  1. Please check English grammar and lexicon. Look for misprints. Several sentences of the manuscript are difficult to follow. Choose simplicity, be not too wordy.

Response: We agree to this point and an authorized person corrected the language.

  1. Please be sure to conform all the aspects of the manuscript to the journal guidelines. I found several discrepancies throughout the text.

Response: We agree to this point and corrected it.

TITLE AND ABSTRACT

  1. The title of the manuscript conveys with the major concern of the study.
  2. The abstract properly summarize the topic addressed. Report data about the main results.

INTRODUCTION

  1. The null hypotheses are incorrectly reported. The statistical null hypothesis assumes that there are not significant differences among the studied groups. Please modify.

Response: We agree with this point and corrected both hypotheses in the manuscript.

MATERIALS AND METHODS

  1. Pag. 4 line 158: table 4, you mean table 3. Please modify.

Response: We agree to this point and corrected it accordingly in the manuscript.

  1. Pag. 5 line 172: which concentration of HF gel did you apply?

Response: We agree to this point and added pieces of information. We etched with 5 %  fluorhidric acid.

RESULTS

  1. Do not duplicate information in the tables and the figures. Choose only one way to report data.

Response: We agree to this point and deleted the table.

DISCUSSION

  1. The null hypotheses are unproperly stated. The null hypothesis must be rejected, not accepted.

Response: We corrected this aspect of the manuscript.

  1. Indications for further research are too weak. Please add others.

Response: We agree to this point and added information about including in the future research other materials such as hybrid, lithium disilicate glass-ceramic.

  1. You should discuss your data in the discussion section, comparing the relevant ones with the ones from similar studies. For example, a very recent study from Mavriqi et al. (1) reported Vickers hardness of ZLS Suprinity to be 7.6 ± 0.7 GPa with no thermal aging. Please explain the differences compared with your study.

 Response: In this study, the samples were glazed using two-layer of the specific glaze. The microhardness was tested using specific equipment with an applied force of 300 g for 10 seconds for each sample. The tests were done in three different areas. We think the glaze layer could be an explanation for the different results before the thermal aging of the samples.

Mavriqi L, Valente F, Murmura G, Sinjari B, Macrì M, Trubiani O, et al. Lithium disilicate and zirconia reinforced lithium silicate glass-ceramics for CAD/CAM dental restorations: biocompatibility, mechanical and microstructural properties after crystallization. Journal of Dentistry. 2022 Apr 1;119:104054.

We added in the manuscript as well the reference because it was helpful.

Round 2

Reviewer 1 Report

Minor corrections:

·        Sometimes you write heat-pressed and sometimes hot-pressed. What is the right way? It should be the same throughout the manuscript.

·        Correct the order of the references throughout the manuscript. It should follow the numbers in the reference section

·        Table 5 should be numbered Table 4

·        Figure 4- why the legend for Figure 4 is deleted?

Author Response

We thank You very much for the review!

Minor corrections:

  • Sometimes you write heat-pressed and sometimes hot-pressed. What is the right way? It should be the same throughout the manuscript.

      Response: We agree to this point and we changed to –hot-pressed- for the whole manuscript. Both terms are accepted and describe the same method of obtaining the glass-ceramic samples.

  • Correct the order of the references throughout the manuscript. It should follow the numbers in the reference section

Response: We agree on this point and corrected the references.

  • Table 5 should be numbered Table 4

      Response: We agree on this point and corrected it.

  • Figure 4- why the legend for Figure 4 is deleted?

      Response: We agree on this point and added the legend.